# Modulation of *I*_Na_, *I*_h_, and *I*_K(erg)_ by Extracellular or Intracellular QX-314 (*N*-(2,6-dimethylphenylcarbamoylmethyl) triethylammonium bromide) in Pituitary Tumor Cells

**DOI:** 10.3390/ijms26178469

**Published:** 2025-08-31

**Authors:** Jeffrey Chi-Fei Wang, Hung-Tsung Hsiao, Sheng-Nan Wu

**Affiliations:** 1Department of Anesthesiology, National Cheng Kung University Hospital, College of Medicine, National Cheng Kung University, Tainan 701, Taiwan; cfwang0505@gmail.com (J.C.-F.W.); aneshsiao@gmail.com (H.-T.H.); 2Department of Physiology, College of Medicine, National Cheng Kung University, Tainan 701, Taiwan; 3Department of Post-Baccalaureate Medicine, National Sun Yat-sen University Medical College, Kaohsiung 804, Taiwan; 4Department of Research and Education, An Nan Hospital, China Medical University, Tainan 70965, Taiwan

**Keywords:** QX-314 (*N*-(2,6-dimethylphenylcarbamoylmethyl) triethylammonium bromide), voltage-gated Na^+^ current, hyperpolarization-activated cation current, *erg*-mediated K^+^ current, Voltage-dependent hysteresis, current kinetics

## Abstract

QX-314 is a positively charged lidocaine derivative with the membrane-impermeant property. This compound applied at the intracellular side has been shown to suppress the voltage-gated Na^+^ current (*I*_Na_), while lidocaine itself acts to suppress the hyperpolarization-activated cation current (*I*_h_). To what extent this drug may exert any effects on various plasmalemmal ionic currents still remains largely unknown. This investigation focused on the impact of QX-314 on ionic currents in GH_3_ cells derived from pituitary tumors. This compound applied extracellularly was noted to differentially suppress the amplitude of transient and late *I*_Na_ with an IC_50_ value of 93 and 42 μM, respectively. In GH_3_ cells dialyzed with QX-314 (10 μM), the *I*_Na(T)_ amplitude evoked by a brief depolarizing step was decreased, and its inactivation was increased. Moreover, QX-314, when applied extracellularly at 100 μM, diminished the amplitude of the *I*_h_ current with an IC_50_ of 68 μM. Intracellular dialysis with QX-314 also suppressed *I*_h_ amplitude; moreover, the later application of oxaliplatin reversed this suppression. As cells were extracellularly and continually exposed to QX-314, the magnitude of the *erg*-mediated K^+^ current (*I*_K(erg)_) was also effectively suppressed with an IC_50_ value of 73 μM. Furthermore, upon intracellular dialysis with QX-314 (10 μM), the degree of the voltage-dependent hysteresis (Hys_(V)_) of *I*_K(erg)_ during the long-lasting isosceles-triangular ramp voltage was decreased; during continued exposure to QX-314, further extracellular bath additions of PD118057 (10 μM) counteracted QX-314-induced suppression. However, the extracellular addition of QX-314 (100 μM) mildly suppressed the outward delayed rectifier K^+^ current in GH_3_ cells. Collectively, QX-314 effectively suppressed *I*_Na_, *I*_h_, and *I*_K(erg)_ in GH_3_ cells, a model of endocrine function, and these actions may contribute to their physiological functions, if similar effects are observed in vivo.

## 1. Introduction

QX-314, a quaternary lidocaine derivative, has been increasingly recognized as a modulator of ionic currents, including voltage-gated Na^+^ current and transient receptor potential currents [1,2,3,4,5,6,7,8]. Previous work has demonstrated the capability of lidocaine to suppress hyperpolarization-activated cation currents in various types of neurons and in heart cells [9,10,11,12]. QX-314 has been demonstrated to produce long-lasting local anesthesia with a slow onset in animal models in vivo [13,14,15]. This compound was also previously reported to suppress the acid-induced activation of esophageal nociceptive C fiber neurons [16].

Nine isoforms of voltage-gated Na^+^ (Na_V_) channels—designated Na_V_1.1 through 1.9 (corresponding to *SCN1A*-*SCN5A* and *SCN8A*-*SCN11A*)—are known to be distributed across various excitable tissues in mammals, including those of the endocrine system [17,18,19]. These Na_V_ channels mediated whole-cell voltage-gated Na^+^ currents (*I*_Na_), which are characterized by a rapid and transient activation phase (transient *I*_Na_ or *I*_Na(T)_), followed by swift inactivation. Despite this inactivation, a small but physiologically significant component of the current persists, referred to as late *I*_Na_ (*I*_Na(L)_) [18,20]. Previous studies have shown that the intracellular application of QX-314 can effectively suppress *I*_Na_ in several types of excitable cells [21,22]. However, the extent to which QX-314—administered either intracellularly or extracellularly—can influence the amplitude and/or gating properties of *I*_Na_ remains largely uninvestigated.

The hyperpolarization-activated cation current (*I*_h_), also known as the funny current (*I*_f_), plays a critical role in regulating rhythmic electrical activity across various excitable cell types, including cardiac cells, neurons, and both neuroendocrine and endocrine cells [23,24,25,26,27,28,29,30,31]. This type of current exhibits mixed Na^+^/K^+^ permeability, reflecting an unusual ion selectivity that permits the conduction of Na^+^ and K^+^ ions alike. It is susceptible to inhibition by CsCl, ZD7288, and ivabradine, and can be activated by oxaliplatin (OXAL) [24,27,32,33,34,35,36]. When activated at resting membrane potential, these currents may generate a net inward current primarily carried by Na^+^, which subsequently depolarizes the membrane potential toward the threshold necessary for triggering an action potential [2,23,28,29,32,35,37,38,39]. These ionic currents are attributed to channels encoded by the hyperpolarization-activated cyclic nucleotide-gated (*HCN*) gene family, a subset of the larger superfamily comprising voltage-gated K^+^ channels and cyclic nucleotide-gated channels. Notably, lidocaine has been shown to attenuate the amplitude of *I*_h_ in both substantia gelatinosa neurons and thalamocortical neurons [10,11,12,40,41]. In contrast, prior investigations demonstrated that intracellular application of QX-314 effectively reduced *I*_h_ in hippocampal neurons [42,43], while its extracellular application failed to elicit a comparable effect [10]. These results point to an ongoing discussion surrounding the mechanism through which QX-314 engages *HCN* channels to alter *I*_h_ amplitude and gating dynamics.

The *ether-à-go-go*-related gene, (*erg*)-mediated K^+^ current (*I*_K(erg)_) is present in a variety excitable cell types, including heart cells, endocrine cells, and cells of the neuroendocrine system [44,45,46,47,48,49]. This current is carried by the *EAG* (*ether-à-go-go*) family voltage-gated K^+^ (K_V_) channels, characterized by distinctive gating behavior—marked by swift inactivation and notably delayed deactivation. These currents play a vital role in maintaining the baseline potential across the cell membrane and in fine-tuning subthreshold responsiveness and rhythmic firing behavior [48,49]. Consequently, it is crucial to investigate how QX-314 influences the amplitude of *I*_K(erg)_, its channel-opening and closing dynamics, and the voltage-driven memory effect known as voltage-dependent hysteresis (Hys_(V)_).

With this background, the research aims to investigate how QX-314, a compound known to block ion channels, impacts the electrical properties of GH_3_ cells. Specifically, the study aims to delineate its impact on several critical transmembrane ionic currents, including the transient *I*_Na_ (*I*_Na(T)_), the late *I*_Na_ (*I*_Na(L)_), the hyperpolarization-activated cation current (*I*_h_), the K^+^ current mediated by *erg* (*I*_k(erg)_), and the voltage-dependent delayed-rectifier K^+^ current (*I*_K(DR)_). The use of GH_3_ cells, a well-established and stable model derived from a pituitary tumor, provides a reliable system for these experiments.

## 2. Results

### 2.1. Inhibitory Effect of QX-314 on the Amplitude of Voltage-Gated Na^+^ Currents (I_Na_) Measured from GH_3_ Cells

The initial phase of the experiment aimed to assess the impact of extracellular QX-314 on the amplitude of transient *I*_Na_ (*I*_Na(T)_) and late *I*_Na_ (*I*_Na(L)_) in these cells. To isolate *I*_Na_, we placed cells in a Ca^2+^-free Tyrode’s solution, supplemented with 10 mM tetraethylammonium chloride (TEA) and 0.5 mM CdCl_2_. TEA or CdCl_2_ was included to inhibit nonspecific K^+^ and Ca^2+^ currents, respectively. Additionally, the recording pipettes were filled with a Cs^+^-enriched internal solution. Once the whole-cell configuration was successfully established, the membrane potential of a recorded cell was held at −80 mV. A 40 ms depolarizing pulse to −10 mV was then applied to evoke *I*_Na_. As anticipated, this voltage step induced a robust inward current characterized by rapid activation and inactivation kinetics. The current was effectively blocked by tetrodotoxin (TTX, 1 μM), confirming its identity as *I*_Na_, whereas nimodipine (1 μM) had no inhibitory effect, indicating minimal contribution from L-type Ca^2+^ channels [50,51,52]. Continued exposure of the cells to QX-314 resulted in a progressive reduction in the amplitude of *I*_Na_, as illustrated in Figure 1A. For example, a bath addition of QX-314 at a concentration of either 100 or 300 μM markedly decreased the *I*_Na(T)_ amplitude from 594 ± 23 pA to either 452 ± 21 pA (*n* = 8, *p* < 0.05) or 312 ± 16 pA (*n* = 8, *p* < 0.05), respectively, while such maneuver, respectively, decreased the I_Na(L)_ amplitude to 76 ± 7 pA (*n* = 8, *p* < 0.05) or 21 ± 5 pA (*n* = 8, *p* < 0.05) from a control value of 92 ± 8 pA (*n* = 8). After washout of QX-314, the amplitudes of *I*_Na(T)_ and *I*_Na(L)_ recovered to 588 ± 21 pA (*n* = 8) and 90 ± 7 pA (*n* = 8), respectively. In addition, exposure to 300 μM QX-314 significantly accelerated the inactivated time course of *I*_Na(T)_, as reflected by a reduction in the slow component of the *I*_Na(T)_-inactivation time constant (τ_inact(S)_) from 7.2 ± 1.3 to 3.9 ± 0.9 ms (*n* = 8, *p* < 0.05). Figure 1B showed that cell exposure to QX-314 could result in a concentration-dependent decrease in the amplitude of *I*_Na(T)_ or *I*_Na(L)_ activated by 40 ms step depolarization. The experimental data indicated that the IC_50_ values for extracellular QX-314-induced inhibition were 93 μM for *I*_Na(T)_ and 42 μM for *I*_Na(L)_, respectively. Our initial measurements, therefore, demonstrated that the extracellular application of QX-314 exerted a suppressive effect on both *I*_Na(T)_ and *I*_Na(L)_ natively expressed in GH_3_ cells. Moreover, the drug showed a greater selectivity toward *I*_Na(L)_ compared with *I*_Na(T)_ activated by short rectangular depolarizing steps.

### 2.2. Effect of Intracellular Dialysis with QX-314 on the Amplitude of I_Na_

Next, we examined whether *I*_Na(T)_ or *I*_Na(L)_ in GH_3_ cells could be affected by intracellular dialysis with QX-314. For these experiments, the recording pipette was filled with a Cs^+^-enriched solution containing 10 μM QX-314. Following establishment of the whole-cell configuration, cells were voltage clamped at −80 mV. A 30 ms hyperpolarizing prepulse to −100 mV was applied every 20 s, followed by a depolarizing step to −10 mV to elicit *I*_Na_. As shown in Figure 2, the intracellular application of 10 μM QX-314 led to a progressive decline in *I*_Na_ amplitude in response to the repetitive depolarizing pulse delivered at a frequency of 0.05 Hz. For instance, immediately following the establishment of the whole-cell configuration, the amplitudes of *I*_Na(T)_ and *I*_Na(L)_ recorded at the first voltage-clamp pulse were 491 ± 31 and 17.3 ± 2.2 pA (*n* = 8), respectively. Notably, these values were significantly reduced by the 20th pulse—approximately 7 min after membrane rupture—with *I*_Na(T)_ and *I*_Na(L)_ declining to 297 ± 19 pA and 3.2 ± 0.3 pA, respectively (*n* = 8, *p* < 0.05 for both). Additionally, the slow inactivation time constant (τ_inact(S)_) of *I*_Na(T)_ decreased markedly from 5.6 ± 0.3 ms at the first pulse to 1.9 ± 0.2 ms at the 20th pulse (*n* = 8, *p* < 0.05). Furthermore, this reduction in *I*_Na(T)_ and *I*_Na(L)_ amplitude observed at the 20th pulse was reversed upon the subsequent application of 10 μM tefluthrin (Tef), a type-I pyrethroid insecticide recently identified as an activator of *I*_Na_ [50,53].

The extracellular and intracellular application of QX-314 resulted in differential inhibition, which is consistent with previous reports demonstrating distinct pharmacokinetics and access routes for the compound [6,7]. The higher concentrations used in bath applications were necessary to overcome the limited membrane permeability of QX-314 and achieve effective inhibition. In contrast, intracellular dialysis shows direct access to the intracellular compartment, enabling robust effects at significantly lower concentrations. We have clarified this rationale in the manuscript to improve transparency.

### 2.3. Inhibitory Effect of QX-314 on the Current Versus Voltage (I-V) Relationship of Hyperpolarization-Activated Cation Currents (I_h_) in GH_3_ Cells

Past research has demonstrated that lidocaine can alter *HCN* currents in substantia gelatinosa neurons and thalamocortical neurons [11,40]. Building on this, we next examined how QX-314 affects *I*_h_ in GH_3_ cells at varying membrane potentials. In these experiments, cells were bathed in a Ca^2+^-free Tyrode’s solution and the recording pipette was filled with K^+^-enriched solution. As demonstrated in Figure 3A,B, in the control period (i.e., neither extracellular nor intracellular QX-314 was present), the currents became progressively increased at a voltage more negative than −80 mV; furthermore, the time course of current activation was noted to become faster with a greater step hyperpolarization, as reported previously [30,33,34,40]. The addition of QX-314 (100 μM) to the bath medium resulted in a reduction in the amplitude of *I*_h_ by responding to 2 s membrane hyperpolarization (Figure 3A). Figure 3B displays the averaged I-V relationship of *I*, which was measured at the conclusion of each hyperpolarizing pulse. The *I*_h_ amplitudes in Figure 3B between the absence and presence of 100 μM QX-314 were significantly distinguishable at the voltages between −80 and −120 mV. Furthermore, the activation time constant (τ_act_) of *I*_h_ at the level of −120 mV was conceivably increased (1.28 ± 0.18 s [control] versus 1.62 ± 0.17 s [in the presence of 100 μM QX-314]; *n* = 8, *p* < 0.05). It is thus important to mention, from these results, that in addition to a reduction in the initial component of *I*_h_, the I-V relationship of sustained *I*_h_ (Figure 3B) became more pronounced. Figure 3C showed that the addition of QX-314 to the bath medium could concentration-dependently decrease the amplitude of *I*_h_ in response to the 2 s membrane hyperpolarization from −40 to −120 mV. According to the modified Hill equation described in the Section 4. the IC_50_ value for QX-314-mediated suppression of the *I*_h_ amplitude in GH_3_ cells was determined to be 68 μM.

### 2.4. Impact of Intracellular Dialysis with QX-314 on the I_h_ Amplitude

Next, we investigated whether intracellular dialysis with QX-314 could alter the *I*_h_ amplitude in response to prolonged membrane hyperpolarization. To isolate *I*_h_, cells were perfused with a Ca^2+^-free Tyrode’s solution supplemented with 1 μM tetrodotoxin (TTX) and 0.5 mM CdCl_2_. The recording electrode was filled with a K^+^-rich internal solution containing 10 μM QX-314. As shown in Figure 4, the intracellular application of 10 μM QX-314 led to a substantial attenuation of the *I*_h_ amplitude. The *I*_h_ amplitude obtained between the 1st pulse and 20th pulse was found to differ vastly. Furthermore, when oxaliplatin (10 μM, OXAL) was subsequently added following the 20th pulse, the *I*_h_ amplitude suppressed by intracellular dialysis with QX-314 was markedly reversed. OXAL, a platinum-based chemotherapeutic drug, was demonstrated to activate *I*_h_ [33].

### 2.5. Suppressive Effect of QX-314 on the Amplitude of erg-Mediated K^+^ Current (I_K(erg)_) in GH_3_ Cells

We also investigated whether extracellular QX-314 modified *I*_K(erg)_ in these cells. To identify *I*_K(erg)_, cells were bathed in a high-K^+^, Ca^2+^-free solution, while the recording pipette was filled with a K^+^-enriched solution. With the cell held at a holding potential of −10 mV, we then applied a series of voltage steps from −110 to −10 mV in 20 mV increments. As depicted in Figure 5A, one minute after cell exposure to 100 μM QX-314, the peak *I*_K(erg)_ was significantly suppressed measured across membrane potentials of −90 and −110 mV. The averaged *I*-*V* relationships for peak and sustained *I*_K(erg)_ are shown in Figure 5B, comparing data from the control condition (without QX-314) to those obtained with 100 μM QX-314 acquired in the control period and during cell exposure to 100 μM QX-314.

### 2.6. Impact of Intracellular Dialysis with QX-314 on Hys_(V)_ of I_K(erg)_ Activated by the Upright Isosceles-Triangular Ramp Voltage (V_ramp_)

The Hys_(V)_ behavior of *I*_K(erg)_, which has been studied in multiple cell types and has been previously characterized in various cell types, is known to significantly influence the electrical properties of diverse, electrically active cells [44,50]. Therefore, we next explored how intracellular dialysis with QX-314 alters the Hys_(V)_ behavior in GH_3_ cells, using a long-lasting isosceles-triangular V_ramp_. In this set of whole-cell current recordings, the cell was held at a membrane potential of −10 mV and subjected to an upright isosceles-triangular V_ramp_ ranging from −110 to 0 mV over 2.4 s (corresponding to a ramp speed of ±0.092 mV/s) delivered via digital-to-analog conversion (Figure 6A). Notably, the current amplitudes elicited by the ascending and descending phases of the triangular V_ramp_ were markedly different. The clear difference between the instantaneous *I*_K(erg)_ and the membrane potential indicates the presence of Hys_(V)_ in the channel’s elicitation, as shown in Figure 6A. Furthermore, when the cell was intracellularly dialyzed with 10 μM QX-314, the *I*_K(erg)_ elicited during the ascending phase of the triangular V_ramp_ was reduced, albeit to a lesser extent than the current observed during the descending phase. For example, in the current trace taken at the first pulse (i.e., immediately after membrane rupture occurred), *I*_K(erg)_s at the level of −60 mV activated by the ascending and descending phases of triangular V_ramp_s were noted to be 197 ± 18 and 87 ± 11 pA (*n* = 8), respectively, the values of which were found to differ significantly (*p* < 0.05). Furthermore, during intracellular dialysis with 10 μM QX-314, subsequent bath application of PD118057 (10 μM) led to a significant reduction in the amplitude of both ascending and descending *I*_K(erg)_s recorded at −60 mV during the 20th pulse, decreasing to 101 ± 17 and 49 ± 7 pA (*n* = 8, *p* < 0.05). PD118057 was previously reported to enhance the *I*_K(erg)_ [51].

### 2.7. Mild Inhibitory Effect of QX-314 on Averaged I-V Relationship of Delayed Rectifier K^+^ Current (I_K(DR)_) Recorded from GH_3_ Cells

Lidocaine at higher concentrations was previously reported to suppress voltage-gated K^+^ currents [52,53]. In another set of measurements, we thus wanted to explore if cell exposure to QX-314 could modify various K^+^ currents, specifically the delayed rectifier K^+^ current (*I*_K(DR)_), in GH_3_ cells. To isolate these currents, GH_3_ cells were perfused with a Ca^2+^-free Tyrode’s solution. The cells were held at a membrane potential of −50 mV under voltage-clamp conditions and subsequently subjected to 1 s depolarizing voltage steps ranging from −50 to +50 mV in 10 mV increments. Under this voltage-clamp protocol, a distinct family of *I*_K(DR)_ was reliably evoked in response to sustained membrane depolarization. These currents exhibited outward rectification and a modest relaxation in their inactivation kinetics. Of note, following a 1 min application of 100 μM QX-314, the *I*_K(DR)_ amplitude exhibited a modest reduction, especially at membrane potentials between +30 and +50 mV (Figure 7).

## 3. Discussion

The results of this study show that the structural features of QX-314 are key to its ability to block *I*_Na_, *I*_h_, and *I*_K(erg)_ when applied extracellularly at high concentrations. As QX-314 was applied on the extracellular side, the IC_50_ value needed for its block of *I*_Na(T)_, *I*_Na(L)_, *I*_h_, or *I*_K(erg)_ observed in GH_3_ cells was optimally estimated to be 93, 42, 68, or 73 μM, respectively, although the *I*_K(DR)_ amplitude was mildly suppressed. Since QX-314 is a membrane-impermeant molecule that does not diffuse through the lipid bilayer [7], it is plausible to assume it acts on these ionic currents via extracellular binding sites. However, its intracellular binding site may have a higher affinity for the specified ionic channels.

In accordance with previous studies, QX-314 could be an intracellular blocker of *I*_Na_ [1,21,22]. However, in our study, intracellular dialysis with QX-314 suppressed the amplitude of *I*_Na(T)_ less than the amplitude of *I*_Na(L)_. Furthermore, adding Tef, a known activator of *I*_Na_ [48,49], attenuated the decrease in the *I*_Na_ magnitude caused by QX-314. QX-314 applied intracellularly was also noted to shorten the τ_inact(S)_ value of *I*_Na(T)_. As a result, the QX-314 molecule might have a greater affinity for the open state of Na_V_ channels than for their closed or resting state in GH_3_ cells [1]. This might explain the slow onset and long-lasting anesthesia observed in animal models [13,14,53].

Research has previously established the suppressive effect of lidocaine on *I*_h_ in various types of neurons and in heart cells [9,10,11,12,39]. In this study, we demonstrated that intracellular QX-314 suppressed the amplitude of *I*_h_ and decreased its activation time constant. The blocking of *I*_h_ by QX-314, applied extracellularly or intracellularly, is likely not instantaneous. Instead, the block develops over time after the *HCN* channel opens, causing a subsequent slowing of the current during QX-314 exposure. With continued dialysis with QX-314, additional application of OXAL, an activator of *I*_h_ [32], could effectively reverse the QX-314-mediated suppression of the *I*_h_ amplitude. In previous studies, *HCN2*, *HCN3*, or mixed *HCN2*+*HCN3* channels have been expressed in GH_3_ cells or other endocrine or neuroendocrine cells [24,31]. Based on the lack of isoform-specific effects of QX-314 on the magnitude and gating of these *HCN*-encoded currents, it is likely that QX-314 inhibits at least three of the four mammalian *HCN* isoforms: *HCN1*, *HCN2*, and *HCN4*. These three isoforms are all found in both neurons and cardiomyocytes [19,23]. Moreover, the QX-314-mediated suppression of *I*_h_ observed in this study would likely affect the spontaneous electrical activity in diverse excitable cell types, such as GH_3_ cells [23,25,36]. It is also reasonable to reflect that the inhibition of *HCN*-encoded currents by QX-314 may be intimately linked to changes in either stimulus–secretion coupling (e.g., exocytosis) in endocrine/neuroendocrine cells, peripheral sensation, or analgesic actions [2,10,15,16,22,24,30].

This study provides evidence that QX-314 applied intracellularly or extracellularly was able to suppress the magnitude of *I*_K(erg)_ evoked by long-lasting membrane hyperpolarization. This work also demonstrated that GH_3_ cells exhibited the dynamically changing Hys_(V)_ behavior of *I*_K(erg)_ when activated by an upright isosceles-triangular V_ramp_, which strongly reflects the remarkable voltage dependence of double V_ramp_-induced *I*_K(erg)_ in GH_3_ cells [50]. It is further important to mention that intracellular dialysis with QX-314 (10 μM) was found to decrease the Δarea of *I*_K(erg)_’s Hys_(V)_. With continued dialysis with QX-314, the further addition of PD118057, an activator of *I*_K(erg)_ [51], was found to counteract QX-314 mediated decrease inf Hys_(V)_’s strength of the current. Consequently, these results strongly suggest that the Hys_(V)_’s degree of *I*_K(erg)_ observed in these cells can be evidently perturbed. It is therefore reasonable to assume that QX-314 applied intracellularly possesses positive charges, which may become the protonated or quaternized forms and in turn exert multiple interactions with negative charge amino acid residues in the pore region and/or voltage-sensing domains of K_erg_ channels, thereby blocking the pore and restricting the movement of the S4 voltage-sensing segments. The QX-314 molecule can enter K_erg_-channel pores from the intracellular side and block K^+^ ion movements through the channel at a depolarized potential. To what extent an QX-314-mediated block of *I*_K(erg)_ (i.e., rapidly activating delayed rectifier K^+^ current) affects cardiac function [43] remains to be further resolved.

QX-314 is known to preferentially block channels in their open state of the Na_V_ channel, and our protocol involved a series of pulses to promote channel opening. However, when QX-314 is applied intracellularly, its access to the channel pore is direct and not dependent on extracellular activation. In this context, the presence of external agonists (e.g., Tef) may alter channel gating kinetics or induce conformational states that reduce QX-314’s binding affinity or accessibility from the intracellular side [48,49]. One possible explanation for the cancelation of the blocking effect is that agonist-induced conformational changes stabilize a channel state that is less susceptible to intracellular QX-314 binding, despite increased open-state probability. Alternatively, agonist binding may promote the rapid inactivation or desensitization of the channel, thereby limiting the time window during which QX-314 can exert its blocking action.

Previous reports have indicated that QX-314 is a membrane-impermeable derivative of lidocaine [7]. However, our findings and those of other studies [39,40,41] have suggested that its site of action is intracellular. How this compound gains access to the cytosol remains unclear. One possible explanation is that it enters the cell via other permeabilizing mechanisms, thereby exerting its inhibitory effects on ionic currents. This unique property has made QX-314 a valuable tool in neuroscience and pain research, especially in efforts to achieve targeted, long-lasting local anesthesia without affecting surrounding tissues.

QX-314 is a quaternary lidocaine derivative that carries a permanent positive charge, which prevents it from passively crossing the lipid bilayer of the neuronal membrane [7]. However, recent studies have demonstrated that QX-314 can selectively enter nociceptive neurons via the transient receptor potential vanilloid 1 (TRPV1) channel, which forms large, cation-permeable pores upon activation by agents such as capsaicin [3]. Once inside the cell, QX-314 binds to the intracellular vestibule of voltage-gated ion channels, thereby modulating the initiation and propagation of action potentials. This blockade results in long-lasting analgesia with minimal motor impairment due to the selective expression of TRPV1 on pain-sensing neurons [3,5].

Lidocaine is a well-established local anesthetic known for its non-selective blockade of Na_V_ channels [2,52]. It inhibits neuronal excitability by stabilizing the neuronal membrane and preventing the initiation and conduction of action potential. In addition to Na_V_ channels, lidocaine has been shown to modulate other ion channels, including transient outward potassium currents. These broader effects contribute to its widespread use but also limit its specificity in mechanistic studies. Therefore, in this study, we focused on QX-314, which—when delivered intracellularly, possibly via TRPV1 channels—offers a more targeted approach to ion channel blockade [2,3,5,6,7,9].

## 4. Materials and Methods

### 4.1. Chemicals, Drugs, Reagents, and Solutions Used in This Study

QX-314 (CAS No.: 21306-56-9; also known as lidocaine *N*-ethyl bromide, N-ethyllidocaine bromide, and *N*-(2,6-dimethylphenylcarbamoylmethyl)triethylammonium bromide purchased from Sigma-Aldrich (Genechain, Kaohsiung, Taiwan); and chemical formula C_16_H_27_BrN_2_O), along with nimodipine, oxaliplatin (OXAL), tefluthrin (Tef), tetraethylammonium chloride (TEA), and tetrodotoxin (TTX) were procured from Sigma-Aldrich (Genechain, Kaohsiung, Taiwan). PD118057 was obtained from Tocris (Union Biomed, Taipei, Taiwan). Unless stated otherwise, horse and fetal calf sera, along with L-glutamine, trypsin/EDTA, and culture media (e.g., Ham’s F-12 medium) were sourced from HyClone^TM^ (Thermo Fisher; Level Biotech, Tainan, Taiwan). Other chemicals and reagents were primarily supplied by Sigma-Aldrich or Merck (Genechain) and were of laboratory grade, acquired from standard, reliable sources.

In this study, the standard extracellular medium consisted of HEPES-buffered normal Tyrode’s solution, formulated with the following ionic compositions (in mM): 136.5 NaCl, 1.8 CaCl_2_, 5.4 KCl, 0.53 MgCl_2_, 5.5 glucose, and 5.5 HEPES, and the pH was brought to 7.4 using NaOH. To examine *I*_h_ or *I*_Na_, a Ca^2+^-free variant of Tyrode’s solution—matching the standard composition but lacking CaCl_2_—was employed.

To examine *I*_K(erg)_, the extracellular medium was replaced with a high-K^+^, Ca^2+^-free bath composition containing (in mM): 130 KCl, 10 NaCl, 3 MgCl_2_, and 5 HEPES, with the pH adjusted to 7.4 using KOH. Patch pipettes used for recording *I*_h_ and *I*_K(erg)_ currents were loaded with an internal solution comprising (in mM): 130 K-aspartate, 20 KCl, 1 KH_2_PO_4_, 1 MgCl_2_, 3 Na_2_ATP, 0.1 Na_2_GTP, 0.1 EGTA, and 5 HEPES. The pH of this solution was brought to 7.2 using KOH. For *I*_Na_, the K^+^ ions in the internal solution were substituted with Cs^+^ ions, and the pH was adjusted to 7.2 using CsOH. In specific experimental protocol, the intracellular recording medium was loaded with 10 μM QX-314 to measure changes in ionic currents. All reagents were formulated using deionized water that had been further refined via the Milli-Q^®^ ultrapure purification system (Merck, Tainan, Taiwan).

### 4.2. Cell Preparation

GH_3_ pituitary tumor cells (BCRC-60015), obtained from the Bioresources Collection and Research Center in Hsinchu, Taiwan, were cultured in Ham’s F-12 medium supplemented with 15% (*v*/*v*) horse serum, 2.5% (*v*/*v*) fetal calf serum, and 2 mM L-glutamine. Cells were maintained at 37 °C in plastic culture flasks with a 50 mL capacity under monolayer conditions within a humidified incubator containing a CO_2_/air mixture at a ratio of 1:19) [46,47]. Cell proliferation was routinely assessed using a colorimetric method based on a tetrazolium salt (WST), conducted in microtiter plates. Absorbance was measured with an ELISA microplate reader (Dynatech, Chantily, VA, USA). Measurements were performed once cultures attained a 50–70% confluence, typically between 5 and 7 days after seeding.

### 4.3. Electrophysiological Assessment via Patch-Clamp Current Recording

On the day of experimentation, GH_3_ cells were gently dissociated using a 1% trypsin-EDTA solution. A small volume of the resulting cell suspension was transferred into a custom-built chamber mounted on the stage of an inverted DM-II phase-contrast microscope (Leica; Major Instruments, Kaohsiung, Taiwan). Cells were maintained at room temperature (20–25 °C) in normal Tyrode’s solution, with ionic compositions as previously described. Before measurements, cells were given time to settle onto the bottom surface of the chamber.

Patch electrodes were crafted from Kimax-51 borosilicate capillaries (#DWK34500-99; Kimble^®^, Merck, Tainan, Taiwan) using a vertically oriented microelectrode puller (PP-83; Narishige, Major Instruments, Taipei, Taiwan). The resulting electrodes featured tip diameters of approximately 1 μm and exhibited resistances between 2 and 4 MΩ. During electrophysiological recordings, each electrode was securely positioned in an airtight holder equipped with a lateral suction port. Electrical continuity with the internal pipette solution was maintained via a silver-chloride wire.

Transmembrane ionic fluxes were monitored using an adapted variant of conventional whole-cell configuration of the patch-clamp technique, which incorporated dynamic adaptive suction—where negative pressure was progressively reduced in accordance with rising seal resistance. This approach, implemented with either an Axoclamp-2B amplifier (Molecular Devices, Sunnyvale, CA, USA) or an RK-400 amplifier (Bio-Logic, Claix, France), has been shown to enhance gigaseal formation and stability during prolonged recordings [33,47].

Once a high-resistance seal (>1 GΩ) was achieved, the cell membrane was disrupted by controlled suction to establish the whole-cell configuration. To eliminate potential offset artifacts arising from differences between pipette and bath solutions, the potential was zeroed immediately prior to seal formation, and all recordings obtained in whole-cell configuration were subsequently adjusted.

### 4.4. Whole-Cell Current Recordings

Electrophysiological signals, including potential and current tracings, were recorded using an Asus ExpertBook laptop (Tainan, Taiwan) and a high-resolution Digidata 1440A interface (Molecular Devices). The Digidata 1440A performed both analog-to-digital (A/D) and digital-to-analog (D/A) conversions, operating at a sampling rate of 10 kHz or more. Data acquisition was controlled by pCLAMP 10.6 software (Molecular Devices) running on a Microsoft Window 7 operating system. Using D/A conversion, the voltage-clamp protocols employing rectangular or ramp voltage (V_ramp_) waveforms were specifically tailored to measure the steady-state current versus voltage (I-V) relationship and voltage-dependent hysteresis (Hys_(V)_) of the target ionic currents specified, such as *I*_K(erg)_.

### 4.5. Data Analyses

To assess the percentage inhibition of the *I*_Na_ amplitude (specifically *I*_Na(T)_ and *I*_Na(L)_) by the extracellular application of QX-314, cells were voltage-clamped at −80 mV. A 40 msec depolarizing pulse to −10 mV was then applied to elicit *I*_Na_. The amplitudes of *I*_Na(T)_ and *I*_Na(L)_ were measured at the onset and end of each depolarizing pulse, respectively, under varying concentrations of QX-314. For *I*_h_ measurements, the cell was held at −40 mV. A 2 s hyperpolarizing step to −120 mV was applied, and the *I*_h_ amplitude was measured at the end of this step. To record *I*_K(erg)_, cells were bathed in a high-K^+^, Ca^2+^-free solution. The membrane potential was set at −10 mV, and a 1 s step hyperpolarization to −110 mV was applied to evoke deactivating *I*_K(erg)_s. The peak current amplitude was measured at the onset of the hyperpolarization pulse.

To evaluate the concentration-dependent inhibition of *I*_Na(T)_, *I*_Na(L)_, *I*_h_, and *I*_K(erg)_ by extracellular QX-314, the data were fitted using a modified Hill equation, specifically the three-parameter logistic equation, expressed as follows:$$\mathrm{Percentage}\;\mathrm{inhibition}\left(\%\right)=\left(E_{max}\times {\left[QX\right]}^{n_{H}}\right)/\left({IC}_{50}^{H}+{\left[QX\right]}^{n_{H}}\right)$$
In this equation, [*QX*] is the extracellular concentration of QX-314. *IC*_50_ is the concentration needed for 50% inhibition. *n_H_* is the Hill coefficient, which indicates the degree of cooperativity in the drug’s effect. Lastly, *E*_max_ represents the maximal inhibition of the ionic current that can be induced by QX-314.

### 4.6. Curve-Fitting Procedures and Statistical Analyses

For linear or nonlinear curve-fitting analyses, we employed a range of computational approaches, including the Microsoft^®^ “Solver” add-in embedded in Excel^TM^ 2022 (Microsoft^®^ 365), as well as the 64-bit OriginPro^®^ 2021 software (OriginLab; Scientific Formosa, Kaohsiung, Taiwan) [48]. Ionic currents were expressed as mean ± standard error of the mean (SEM), with sample sizes (n) representing the number of cells from which independent measurements were obtained. All graphical results were presented as mean ± SEM.

To assess statistical differences between two groups, we initially applied either paired or unpaired Student’s *t*-tests. For comparisons involving more than two groups, we conducted a one-way or two-way analysis of variance (ANOVA), followed by Duncan’s multiple range post hoc tests. Formal assessments of data normality were conducted using the Shapiro–Wilk test across all relevant datasets. The results demonstrated no significant deviation from a normal distribution, with *p*-values exceeding 0.05 for all groups. Statistical significance was defined as *p* < 0.05, and significant differences are indicated in the figures by asterisks (* or **).

## 5. Conclusions

The major findings in this study demonstrate seven key effects of the compound QX-314 on ionic currents. First, when applied to GH_3_ cells from the outside, QX-314 was more effective at suppressing *I*_Na(L)_ than *I*_Na(T)_ amplitudes. Second, the intracellular application of QX-314 (10 μM in the patch pipette) caused a progressive decrease in the *I*_Na_ amplitude and accelerated its inactivation; moreover, this suppressive effect on *I*_Na_ was partially reversed by the presence of Tef. Third, as QX-314 was applied extracellularly, the amplitude of *I*_h_ in response to long-lasting membrane hyperpolarization was suppressed with considerable slowing in the activation time course of the current (i.e., an increase in τ_act_ value). Fourth, in cells internally dialyzed with QX-314, the *I*_h_ amplitude responding to the long-lasting hyperpolarizing step was progressively suppressed. This inhibition was subsequently reversed by the addition of oxaliplatin (OXAL). Fifth, with the presence of extracellular QX-314, the *I*_K(erg)_ amplitude activated by membrane hyperpolarization was overly depressed. Sixth, upon intracellular dialysis with QX-314, the *I*_K(erg)_ measured at the entire voltage-clamp level was decreased, and, with continued dialysis with QX-314, the additional presence of PD118057, a stimulator of *I*_K(erg)_, reversed the QX-314-induced decrease in *I*_K(erg)_ amplitude. Seventh, the amplitude of *I*_K(DR)_, in response to membrane depolarization, was slightly decreased with the extracellular presence of QX-314.

The differential effects of QX-314 via extracellular and intracellular routes suggest potential strategies for the targeted modulation of neuronal excitability, which may be relevant for developing localized analgesic or anti-epileptic interventions. Furthermore, the presence of QX-314, whether inside or outside the cell, may alter the magnitude or gating of *I*_Na_, *I*_h_, and *I*_K(erg)_. This may lead to the disruption of electrophysiological activity in diverse excitable cells, a phenomenon observable in both in vitro culture models and in vivo biological systems [5,7,41,53].

## Figures and Tables

**Figure 1 ijms-26-08469-f001:**
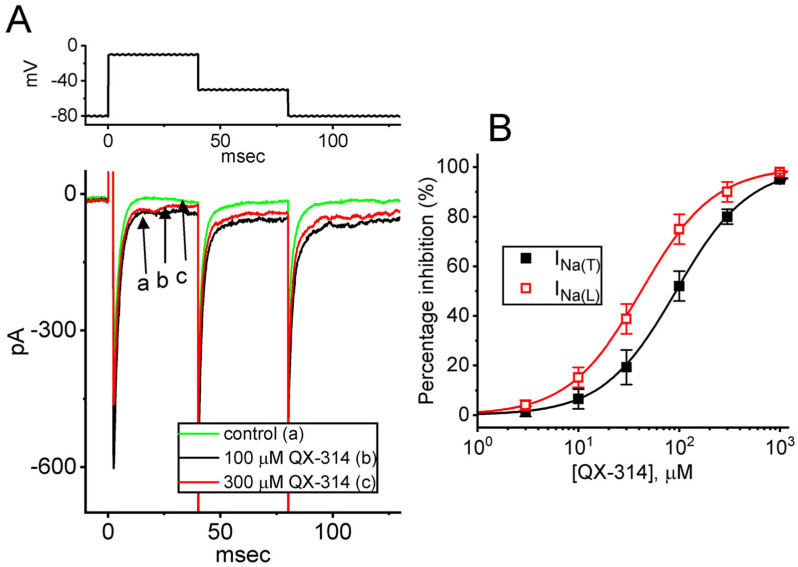
Inhibitory effect of QX-314 on voltage-gated Na^+^ current (*I*_Na_) identified from GH_3_ cells. In this experimental series, cells were dissociated and subsequently maintained in a Ca^2+^-free Tyrode’s solution supplemented with 10 mM tetraethylammonium chloride (TEA) and 0.5 mM CdCl_2_. For electrophysiological recordings, the patch pipette was filled with a Cs^+^-enriched internal solution. (**A**) Representative *I*_Na_ traces evoked by a 40 ms depolarizing pulse from −80 to −10 mV (indicated in the upper part). Current trace labeled “a” is the control (i.e., absence of QX-314, black color), and those labeled “b” or “c” were obtained during cell exposure to 100 μM QX-314 (red color) or 300 μM QX-314 (green color). The recordings were obtained from the same cell to ensure consistency across different concentrations of QX-314 and to minimize variability due to cell-to-cell differences. (**B**) Concentration-dependent inhibition of transient *I*_Na_ (*I*_Na(T)_, black filled squares) and late *I*_Na_ (*I*_Na(L)_, red open squares) by different QX-314 concentrations (mean ± SEM; *n* = 7). Each continuous line was well fit using the modified Hill equation described in the Section 4 ( Materials and Methods). Notably, the addition of QX-314 to the bath suppressed *I*_Na(T)_ and *I*_Na(L)_ with differing potencies.

**Figure 2 ijms-26-08469-f002:**
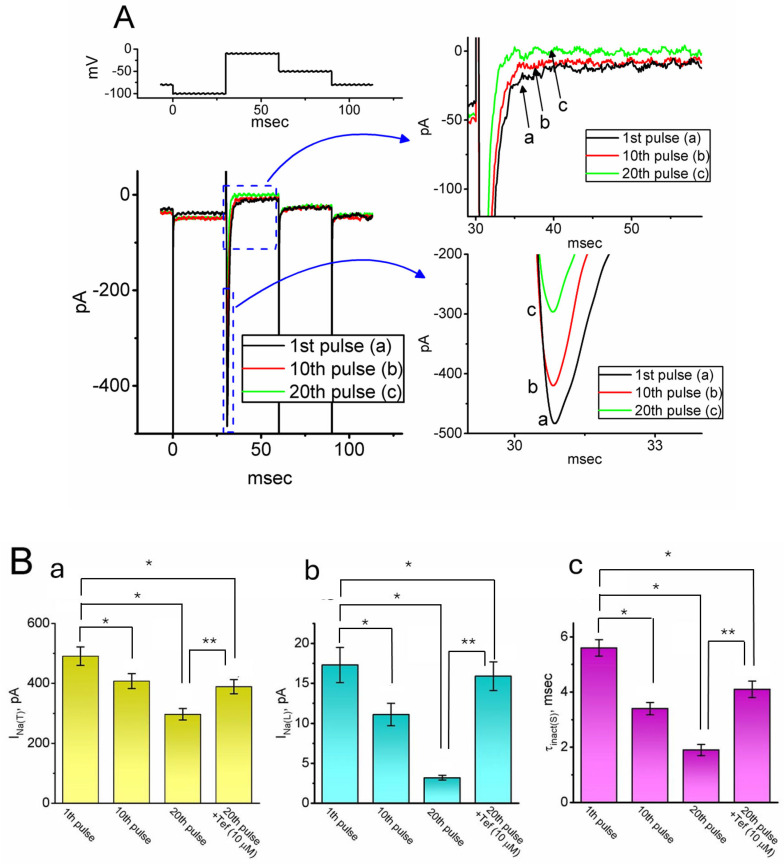
Impact of intracellular dialysis with QX-314 on *I*_Na_ identified in GH_3_ cells. For these experiments, the cells were bathed in a modified Tyrode’s solution devoid of Ca^2+^, supplemented with 10 mM TEA and 0.5 mM CdCl_2_. The patch-clamp recording pipette was filled with an internal solution containing Cs^+^. In (**A**), once whole-cell configuration was successfully achieved, the membrane potential was held at −80 mV. A brief 30 ms hyperpolarizing step to −100 mV was then applied, followed by a 30 ms depolarizing pulse to −10 mV to evoke *I*_Na_, as illustrated in the upper portion of (**A**). In the cell dialyzed with 10 μM QX-314, the trace designated “a” (black color) represents the *I*_Na_ response elicited by the first depolarizing pulse, applied immediately following membrane rupture. The traces marked “b” (red color) and “c” (green color) correspond to recordings obtained during the 10th and 20th depolarizing pulses, respectively. The graph appearing on the right side indicates the expansion recorded from the blue dashed box with a curved arrow. (**B**) Summary bar graph demonstrating effect of intracellular dialysis with 10 μM QX-314 on the amplitude of *I*_Na(T)_ (**a**) and *I*_Na(L)_ (**b**), as well as on the slow component of inactivation time constant (τ_inact(S)_) (**c**) of *I*_Na(T)_ (mean ± SEM; *n* = 8 for each bar with yellow, blue or purple colors). QX-314 (10 μM) was added to the pipette solution. * Significantly different from current amplitude at the 1st pulse (*p* < 0.05), and ** significantly different from that at the 20th pulse (*p* < 0.05).

**Figure 3 ijms-26-08469-f003:**
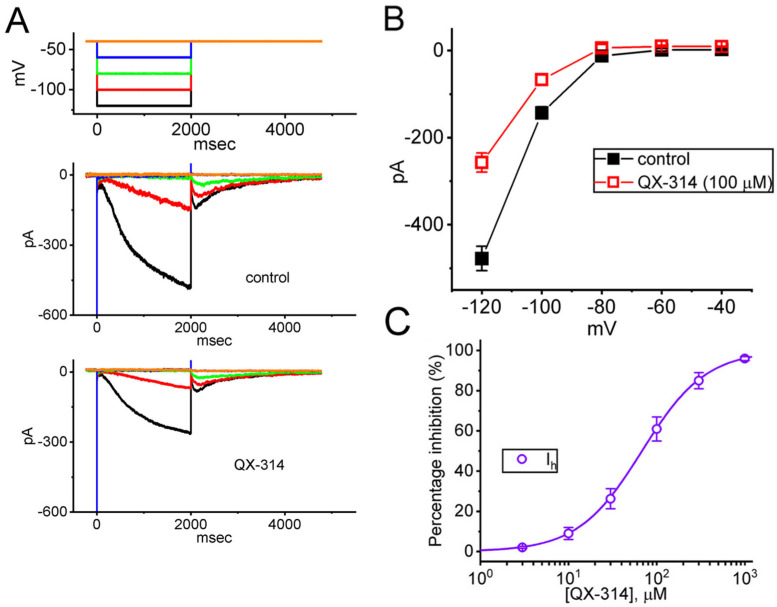
Effect of QX-314 on hyperpolarization-activated cation current (*I_h_*) recorded from GH_3_ cells. Measurements were conducted in cells perfused with a Ca^2+^-free Tyrode’s solution, which contained 1 μM tetrodotoxin (TTX) and 0.5 mM CdCl_2_. The recording electrode was filled with a K^+^-based internal solution, and QX-314 was applied extracellularly to the target cell. During whole-cell current recordings, the cell was held at −40 mV and a series of voltage pulses ranging from −120 to −40 in 20 mV increments was thereafter applied. (**A**) Representative *I*_h_ traces recorded during the control period (absence of QX-314, upper panel) and following exposure to 100 μM QX-314 (lower panel). Voltage-clamp protocol applied is shown at the top of the figure. Current traces are color-coded to match the corresponding potential traces shown in the top part of (**A**). (**B**) Averaged current versus voltage (*I-V*) relationship of *I*_h_ taken without (black filled squares) or with the presence (red open squares) of 100 μM QX-314 (mean ± SEM; *n* = 8 for each point). The studied cell was held at −40 mV and current amplitude was measured at the end of each voltage command applied. (**C**) Concentration-dependent inhibition of *I*_h_ by varying concentrations of QX-314, presented as mean ± SEM (*n* = 7 for each point). *I*_h_ amplitude was quantified at the end of a 2 s hyperpolarizing voltage step from a holding potential of −40 mV to −120 mV. The smooth line illustrates a least squares fit of the data to a modified Hill equation, with fitting parameters and methodological details provided in the Section for Materials and Methods.

**Figure 4 ijms-26-08469-f004:**
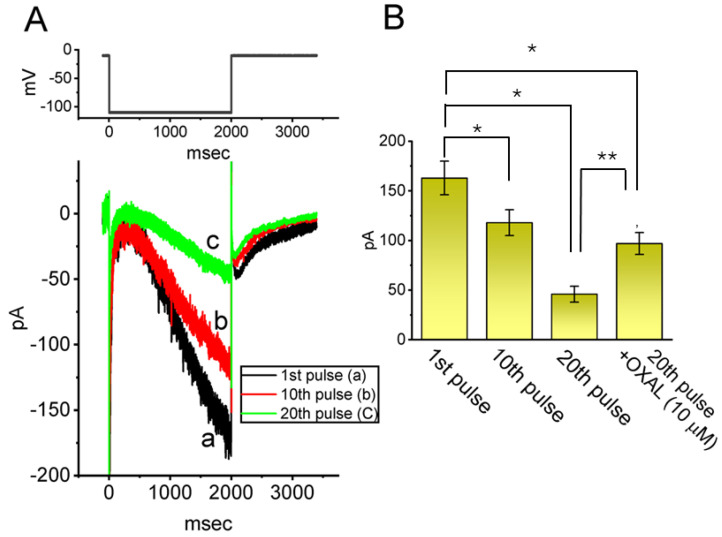
Effect of intracellular dialysis with QX-314 on the amplitude of the hyperpolarization-activated cation current (*I*_h_) identified from GH_3_ cells. To isolate and measure *I*_h_, cells were perfused with a modified Tyrode’s solution devoid of Ca^2+^, supplemented with 1 μM TTX and 0.5 mM CdCl_2_. The internal solution within the recording electrode was enriched with K^+^ ions and contained 10 μM QX-314. In (**A**), upon achieving whole-cell configuration, a 2 s hyperpolarizing voltage step to −110 mV was applied from a holding potential of −40 mV. The top panel shows the applied voltage-clamp protocol. In the cell internally dialyzed with 10 μM QX-314, the current trace labeled “a” (black color) corresponds to the response elicited by the first hyperpolarizing pulse applied immediately following membrane rupture and establishment of whole-cell configuration. In contrast, traces labeled “b” (red color) and “c” (green color) represent recordings obtained during the 10th and 20th successive pulse, respectively, reflecting progressive changes in *I*_h_ as intracellular dialysis proceeded. (**B**) Summary bar graph demonstrating effect of intracellular dialysis with 10 μM QX-314 on the amplitude of *I*_h_ (mean ± SEM; *n* = 8 for each green bar). Current amplitude was measured at the end of a 2 s hyperpolarizing voltage command to −110 mV from a holding potential of −40 mV. OXAL: 10 μM oxaliplatin. * Significantly different from current amplitude at the 1st pulse (*p* < 0.05), and ** significantly different from that at the 20th pulse (*p* < 0.05).

**Figure 5 ijms-26-08469-f005:**
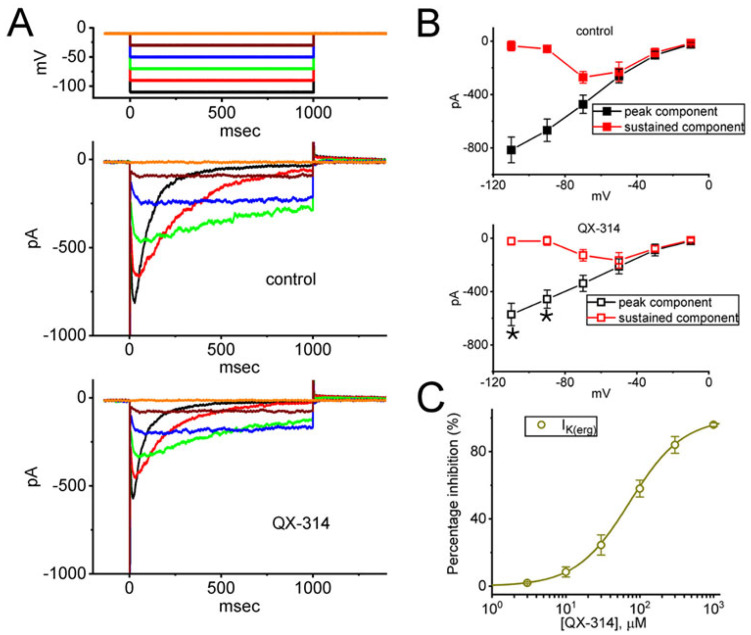
Effect of QX-314 on *erg*-mediated K^+^ current (*I*_K(erg)_) measured from GH_3_ cells. These experiments were conducted in cells which were placed in a high-K^+^, Ca^2+^-free solution, and the measuring electrode was filled with K^+^-enriched solution. During whole-cell current recordings, the examined cell was voltage-clamped at −10 mV, and a series of voltage pulses ranging between −110 and −10 mV in 20 mV steps for a duration of 1 s was delivered to evoke *I*_K(erg)_. (**A**) Representative potential (uppermost part) and current (middle and lower parts) traces. The top panel shows current traces from the untreated cells (without QX-314), while the bottom panel displays traces recorded after the cell was exposed to 30 μM QX-314. Current traces are color-coded to match the corresponding potential traces shown in the top part of (**A**). In (**B**), averaged current–voltage (*I*-*V*) relationships of the peak (black color) and sustained (red color) components of *I*_K(erg)_ are shown, with data from the control condition (without QX-314) in the upper panel and data from the presence of 30 μM QX-314 in the lower panel. All data points are presented as mean ± SEM. The amplitude in peak or sustained components of *I*_K(erg)_ was measured at the start or end of hyperpolarizing voltage command, respectively. Asterisk indicates a statistically significant difference (*p* < 0.05) in the peak deactivating *I*_K(erg)_ (black color) between the absence and presence of 30 μM QX-314 at the membrane potentials of −90 or −110 mV. (**C**) Concentration-dependent inhibition of *I*_K(erg)_ caused by different QX-314 concentrations (mean ± SEM; *n* = 7). To measure the deactivating *I*_K(erg)_, the membrane was hyperpolarized for 1 s from −10 to −110 mV. The current amplitude was recorded at the start of this hyperpolarizing pulse. The solid line represents a curve generated by applying a least squares fitting procedure to the experimental data, using a modified form of the Hill equation. The specific parameters and fitting methods are described in detail in the Section 4. (Materials and Methods). This analytical approach was employed to characterize the relationship between the measured variables and to provide a quantitative model that best represents the observed data trends.

**Figure 6 ijms-26-08469-f006:**
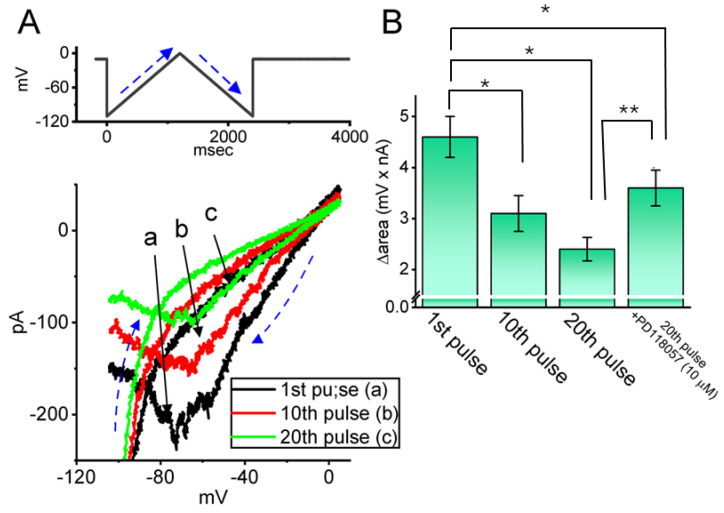
Effect of intracellular dialysis with QX-314 on the voltage-dependent hysteresis (Hys_(V)_) of *I*_K(erg)_ activated in response to isosceles-triangular V_ramp_ recorded from GH_3_ cells. In this set of measurements, we placed cells in a high-K^+^, Ca^2+^-free solution, while the recording electrode was filled with an internal solution which contained 10 μM QX-314. In (**A**), after whole-cell recording was established, we applied a 2.4 s isosceles-triangular V_ramp_ that went from −110 to 0 mV. This ramp was applied every 20 s at a speed of 0.092 mV/s. The upper part in (**A**) denotes the protocol applied. In the cell dialyzed with 10 μM QX-314, the current trace labeled “a” (black color) is the one that was obtained when the first pulse was applied (i.e., immediately after membrane rupture occurred), while those labeled either “b” (red color) or “c” (green color) were obtained at the 10th or 20th pulse, respectively. Blue dashed arrow in (**A**) indicates potential or current trajectory over which time passed during elicitation of a double V_ramp_. (**B**) Quantitative bar graph demonstrating the impact of intracellular dialysis with 10 μM QX-314 on the Δarea of *I*_K(erg)_’s Hys_(V)_ (mean ± SEM; n = 8 for each green bar). The Δarea was calculated from the region enclosed by the curve of *I*_K(erg)_’s Hys_(V)_, which was activated during both the rising and falling phases of a 2.4 s triangular V_ramp_. * Significantly different from V_ramp_-evoked Δarea appearing in the first pulse (*p* < 0.05), and ** significantly different from that in the 20th pulse (*p* < 0.05).

**Figure 7 ijms-26-08469-f007:**
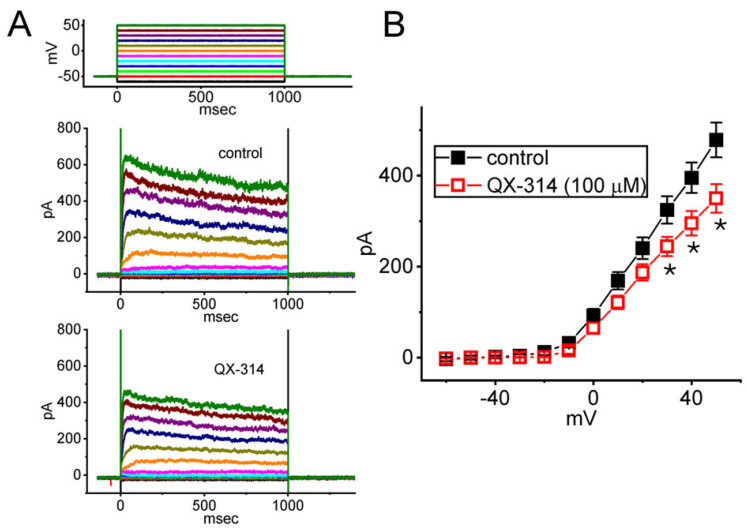
Impact of QX-314 on the delayed rectifier K^+^ current (*I*_K(DR)_) observed in GH_3_ cells. To isolate *I*_K(DR)_, cells were perfused with a Ca^2+^-free Tyrode’s solution supplemented with 1 μM TTX and 0.5 mM CdCl_2_. The patch pipette was filled with a K^+^-enriched solution. Beginning at a holding potential of −50 mV, 1 s voltage pulses ranging from −60 to +50 mV in 10 mV increments were delivered to the cell, as illustrated in the top panel of (**A**). (**A**) Representative *I*_K(DR)_ traces obtained in the control period (i.e., QX-314 was not present, upper part) and during cell exposure to 100 μM QX-314 (lower part). Current traces appearing in different colors correspond with potential traces applied in the uppermost part. Current traces are color-coded to match the corresponding potential traces shown in the top part of (**A**). (**B**) Averaged *I*-*V* relationship of *I*_K(DR)_ obtained without (black filled squares) or with (red open squares) the application of 100 μM QX-314 to the bath. Current amplitude corresponding to each membrane potential was quantified at the termination of the 1 s depolarizing pulse. Each data point represents the mean ± SEM (*n* = 8). An asterisk denotes statistically significant differences (*p* < 0.05) between data points recorded at identical membrane potentials in the presence and absence of QX-314.

## Data Availability

The original contributions presented in this study are included in the article. Further inquiries can be directed to the corresponding author.

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
