# Peer review of "Modulation of INa, Ih, and IK(erg) by Extracellular or Intracellular QX-314 (N-(2,6-dimethylphenylcarbamoylmethyl) triethylammonium bromide) in Pituitary Tumor Cells"

_ijms, 2025, doi:10.3390/ijms26178469_

Round 1

Reviewer 1 Report

Comments and Suggestions for Authors

In this manuscript, Wang et al. investigated the effect of QX-34 on Na, hyperpolarization, and erg-currents in GH3 cells. I have one major and one minor concerns on this manuscript. 

The authors presented a variety of data investigating the effects of QX-314 when applied intracellularly. Given the well establised fact that QX-314 is membrane-impreable, the implications from these results remain unclear, and the authors did not discuss this in further detail. Moreover, the rationale and implications of usage of pituitary cancer GH3 cells are not sufficiently introduced and discussed. 

Minor points: The author should modify the title to be more concise and include the fact that tumor cells are used in this study.

Author Response

Dear You Reviewer Professor:

Reviewer 2 Report

Comments and Suggestions for Authors

The work studies the effect of the lidocaine derivative QX-314 on the activity of voltage-gated sodium and potassium channels as well as HCN channels in GH3 cells. The manuscript requires significant revision.

  1. "We investigated the effects of QX-314 and other relevant compounds on ionic currents." The authors studied only QX-314, not a whole series of lidocaine derivatives. The phrase is incorrect.
  2. The authors study only one compound, which is a derivative of the well-known lidocaine, but they do not specify what makes this substance remarkable and why it should be studied specifically. Thousands of such compounds can be obtained, but it is impossible to study each one. If a single substance is being studied rather than a series, the choice must be justified.
  3. It is necessary to specify the mechanism of action of QX-314, namely the interaction of the substance with channels and the principle of blockade.
  4. What effect does lidocaine itself have on the studied currents? Were experiments conducted with it? The authors briefly mention its non-selective action, but this is insufficient.
  5. The authors did not test samples for normality but used parametric tests and mean values. Without normality tests of distribution, this is improper.
  6. The figures show representative recordings from single cells. In the case of extracellular addition of QX-314, different concentrations are shown (Figure 1). Is this the same cell?
  7. To demonstrate significant differences, it is necessary to use brackets, since it is not clear from the figures in relation to which groups the comparison of the effects of tefluthrin, oxliplatin, PD118057 was carried out. From the figure captions and text, it is not entirely clear how the experiment with tefluthrin, oxaliplatin, PD118057 was conducted: were these substances given immediately after achieving whole-cell or later (after the twentieth pulse)? "and ** significantly different from that in 10th pulse" - why is the comparison in the case of activator addition made with the tenth pulse if assessment was made after the twentieth?
  8. Why do the QX-314 concentrations used in bath application and intracellular dialysis differ by more than an order of magnitude?
  9. The discussion needs to be expanded, as the observed effects are insufficiently compared with literature data. If the authors discovered a side effect, it is necessary to compare results obtained using QX-314 and comment on possible discrepancies explained by the data presented in this article.
  10. The conclusion represents a bullet-point listing of all obtained results. It is necessary to indicate specific prospects for applying the obtained data.
  11. In Figure 5B, the data should be displayed either as diagrams or, at minimum, combined on one graph showing peak component curves for control and QX-314. The same should be done for sustained component curves. In the current form, the differences are not obvious.
  12. The data presented in Figure 7B should be displayed as diagrams to characterize the difference.
  13. One of the main questions: if channel activators are given to the external medium and promote channel opening, and QX-314 acts as a blocker (the authors note "the QX-314 molecule might have a greater affinity toward the open state," and also give not one pulse but a series to open channels), how is the cancellation of the effect in the presence of agonists explained if QX-314 is inside the cell in this case?
Comments on the Quality of English Language

The text requires the revision and has to be corrected by a professional translator. 

Author Response

Dear You Reviewer, Professor:

Round 2

Reviewer 2 Report

Comments and Suggestions for Authors

The authors have addressed my comments.